# Neurophysiological and Clinical Effects of Upper Limb Robot-Assisted Rehabilitation on Motor Recovery in Patients with Subacute Stroke: A Multicenter Randomized Controlled Trial Study Protocol

**DOI:** 10.3390/brainsci13040700

**Published:** 2023-04-21

**Authors:** Sanaz Pournajaf, Giovanni Morone, Sofia Straudi, Michela Goffredo, Maria Rosaria Leo, Rocco Salvatore Calabrò, Giorgio Felzani, Stefano Paolucci, Serena Filoni, Andrea Santamato, Marco Franceschini

**Affiliations:** 1IRCSS San Raffaele Roma, 000163 Rome, Italy; 2Department of Life, Health and Environmental Sciences, University of L’Aquila, 67100 L’Aquila, Italy; 3San Raffaele Istitute of Sulmona, 67039 Sulmona, Italy; 4Department of Neuroscience and Rehabilitation, University of Ferrara, 44121 Ferrara, Italy; 5Department of Neuroscience and Rehabilitation, Ferrara University Hospital, 44121 Ferrara, Italy; 6Villa Bellombra Hospital, 40132 Bologna, Italy; 7IRCCS Centro Neurolesi Bonino-Pulejo, 98124 Messina, Italy; 8IRCCS Fondazione Santa Lucia, 00124 Rome, Italy; 9Fondazione Centri di Riabilitazione Padre Pio Onlus, San Giovani Rotondo, 71013 Foggia, Italy; 10Physical Medicine and Rehabilitative Unit-Riuniti Hospital, University of Foggia, 71100 Foggia, Italy; 11Department of Human Sciences and Promotion of the Quality of Life, San Raffaele University, 00166 Rome, Italy

**Keywords:** stroke, upper limb functions, rehabilitation, robot-assisted therapy, EEG, neuroplasticity

## Abstract

Background: The efficacy of upper limb (UL) robot-assisted therapy (RAT) on functional improvement after stroke remains unclear. However, recently published randomized controlled trials have supported its potential benefits in enhancing the activities of daily living, arm and hand function, and muscle strength. Task-specific and high-intensity exercises are key points in facilitating motor re-learning in neurorehabilitation since RAT can provide an assisted-as-needed approach. This study aims to investigate the clinical effects of an exoskeleton robotic system for UL rehabilitation compared with conventional therapy (CT) in people with subacute stroke. As a secondary aim, we seek to identify patients’ characteristics, which can predict better recovery after UL-RAT and detects whether it could elicit greater brain stimulation. Methods: A total of 84 subacute stroke patients will be recruited from 7 Italian rehabilitation centers over 3 years. The patients will be randomly allocated to either CT (control group, CG) or CT plus UL-RT through an Armeo^®^Power (Hocoma AG, CH, Volketswil, Switzerland) exoskeleton (experimental group, EG). A sample stratification based on distance since onset, DSO (DSO ≤ 30; DSO > 30), and Fugl–Meyer Assessment (FM)-UL (FM-UL ≤ 22; 22 < FM-UL ≤ 44) will be considered for the randomization. The outcomes will be recorded at baseline (T0), after 25 + 3 sessions of intervention (T1), and at 6 months post-stroke (T2). The motor functioning assessed by the FM-UL (0–66) will be considered the primary outcome. The clinical assessments will be set based on the International Classification of Function, Disability and Health (ICF). A patient satisfaction questionnaire will be evaluated in the EG at T1. A subgroup of patients will be evaluated at T0 and T1 via electroencephalography. Their brain electrical activity will be recorded during rest conditions with their eyes closed and open (5 min each). Conclusion: The results of this trial will provide an in-depth understanding of the efficacy of early UL-RAT through a whole arm exoskeleton and how it may relate to the neural plasticity process. The trial was registered at ClinicalTrial.gov with the registration identifier NCT04697368.

## 1. Introduction

Upper limb impairments occur in up to 85% of stroke survivors and may persist even after 6 months from the acute event in 55% to 75% of patients [1]. Among the strategies and recent trends in stroke rehabilitation, robotic technologies are garnering more interest and their applications in clinical routine are growing. Upper limb robot-assisted therapy (UL-RAT) has been proven to improve recovery of arm and hand ranges of motion and muscle strength, thus improving the ability to perform activities of daily living (ADL) [2].

Current opinions about neurorehabilitation strategies to facilitate re-learning and motor recovery have highlighted the importance of task-specific exercises with a high level of intensity and repetition [3]. Indeed, more than 1000 repetitions of upper limb exercise are needed to elicit neuroplastic modifications to address effective post-brain injury recovery [4]. In this context, UL-RAT would be able to provide patients with an assist-as-needed approach.

Although several national guidelines advise clinicians on the importance of UL-RAT for arm stroke recovery, its efficacy has not been totally established, with an important recent trial that failed to demonstrate the superiority of robotic training over intensive conventional arm training [5]. In particular, UL-RAT should have a different efficacy on arm recovery following different subjects’ characteristics, for example, those linked to basal impairment, time from stroke, and subject participation [6]. Indeed, more severely impaired subjects should have more chances to perform intensive and effective training than those that only carry out conventional training [7,8]. On the contrary, less affected subjects have the possibility to perform, even without a robot, adequate arm training in terms of dose and task specificity. Following these considerations, there is the need for a well-powered randomized controlled study (RCT) that identifies subject characteristics for exoskeleton-based UL-RAT [9,10] in subjects affected by stroke in their subacute phase.

Over the last few decades, there has been growing literature demonstrating that the brain is capable of substantial structural and functional reorganization after stroke. Indeed, although some recovery is known to occur spontaneously within the first months, there is growing evidence that neurorehabilitation, with regard to innovative approaches, may modulate neuroplasticity mechanisms, providing greater functional benefits in a larger population of stroke survivors [11]. To investigate the neurophysiological basis of recovery after brain damage, EEG could be of help [12]. Moreover, monitoring brain activity immediately after a stroke or during neurosurgery may be fundamental to achieving better outcomes and prognosis [13,14].

Recent data suggest that the most promising EEG quantifiers for predicting UL motor outcome following stroke are event-related measures [15], although spectral power analysis in some relevant bands and fractal dimension analysis are also promising [16].

The first aim of this RCT is to investigate the clinical effects in terms of improvement in the motor control of an exoskeleton-based UL-RAT added onto standard rehabilitation compared with the same amount of conventional arm training alone.

The secondary objectives of the study are (a) to evaluate the efficacy of UL-RAT on arm ability recovery and on subjects’ quality of life; (b) to evaluate its efficacy in terms of objectives robot measures; (c) to identify the characteristics of patients who may benefit more from UL-RAT in terms of age, onset admission interval, and upper limb impairment; and (d) to evaluate of the effects of UL-RAT on motor recovery measured by neurophysiological signals: muscle and neural modification measured by surface electromyography and high-density electroencephalography (in a subgroup of patients).

## 2. Materials and Methods

### 2.1. Study Design

This trial has been designed as a multicenter, single-blind (evaluator) longitudinal parallel group with stratified block randomization, according to the updated guidelines for reporting parallel group randomized trials (CONSORT) [17].

This RCT has been approved by the Ethical Committee (unique protocol ID: RP 20/08; protocol code: PowerUPS-REHAB) and registered at clinicalregistration.gov (ID: NCT04697368; accessed on 1 March 2023).

### 2.2. Participants (Recruitment)

Inclusion criteria: age between 18 and 85 years old; first-ever ischemic stroke verified by brain imaging (CT or MRI); severe or moderate upper limb hemiparesis according to their Fugl–Meyer Assessment Upper Limb (FM-UL) score (FM-UL ≤ 22; between 22 and 44, respectively); stroke in the subacute phase (≤60 days from the distance since onset—DSO); Modified Ashworth Scale (MAS) of the shoulder, elbow, and wrist < 3; and sufficient cognitive and linguistic level to understand the instructions and to provide informed consent to participate in the study.

Exclusion criteria: stroke located in the brain stem or cerebellum; unstable general clinical conditions; severe visual impairment; recent upper limb botox injection or an injection planned for the duration of the study; orthopedic or neurological disease altering paretic upper limb function; interruption of treatments (both conventional and UL-RAT) for 1 week or 5 consecutive sessions; and participation in other innovative treatment protocols for the upper limb.

### 2.3. Sample Population and Randomization

Eighty-four subjects will be recruited from seven Italian inpatient rehabilitation centers over three years and randomly assigned into one of the following groups: experimental group (EG) receiving UL-RAT or control group (CG) receiving conventional treatment (CT). A sample stratification based on DSO (DSO ≤ 30; DSO > 30) and motor impairment (FM-UL ≤ 22; 22 < FM-UL ≤ 44) will be considered for the randomization.

The patients will be randomly assigned to one of the two groups using a web-based application for block randomization (www.randomization.com; accessed on 6 November 2022). In particular, we will use the block randomization method (block size = 4) in order to ensure balance in the sample sizes across groups over time. Regarding the masking description, the outcome assessor will be blind to the study protocol. Please see the consort flow diagram in Figure 1.

### 2.4. Outcome Assessment

The subjects of both groups will be assessed clinically (intention to treat) by a physical therapist not involved in other phases of the study at baseline (T0), at the end of the treatment (T1), and at 6 months after the acute event (T2). See Figure 1.

### 2.5. Clinical Outcome Measure

#### 2.5.1. Primary Outcome

The FM-UL motor evaluation (0–66) will be considered the primary outcome of the study. The Fugl–Meyer Assessment is a stroke-specific, performance-based impairment index. It is designed to assess motor functioning, balance, sensation, and joint functioning in patients with post-stroke hemiplegia. It is applied clinically and in research to determine the disease severity, to describe motor recovery, and to plan and assess the treatment. In this study, we will consider the motor performance items of the upper extremity (0–66) only, i.e., the FM-UL.

#### 2.5.2. Secondary Outcomes

The secondary outcomes will be based on the ICF:Body function: MAS (shoulder, elbow, and wrist);Activity (capacity/performance): Box and Block Test; Nine Hole Peg Test; Frenchay Arm Test; Modified Barthel Index;Participation: Modified Rankin Scale.

A patient satisfaction questionnaire will be evaluated only in EG at T1.

A change in the spasticity (shoulder, elbow, and wrist) will be measured by the MAS, which measures resistance during passive soft-tissue stretching. Scoring: 0, no increase in muscle tone; 1, a slight increase in muscle tone, manifested by a catch and release or by minimal resistance at the end of the range of motion when the affected part(s) are moved in flexion or extension; 1+, a slight increase in muscle tone, manifested by a catch, followed by minimal resistance throughout the remainder (less than half) of the ROM; 2, a more marked increase in muscle tone through most of the ROM, but affected part(s) easily moved; 3, Considerable increase in muscle tone, passive movement difficult; and 4, affected part(s) rigid during flexion or extension.

Unilateral gross manual dexterity will be measured by the Box and Block Test (BBT). It is a quick, simple, and inexpensive test. The BBT is composed of a wooden box divided into two compartments by a partition and 150 blocks. The BBT administration consists of asking the client to move, one by one, the maximum number of blocks from one compartment of a box to another of equal size within 60 s. The box should be oriented lengthwise and placed at the client’s midline, with the compartment holding the blocks oriented towards the hand being tested. In order to practice and to register baseline scores, the test should begin with the unaffected upper limb. Additionally, a 15 s trial period is permitted at the beginning of each side. Before the trial but after standardized instructions are given to the patients, they should be advised that their fingertips must cross the partition when transferring the blocks and that they do not need to pick up any blocks that might fall outside of the box.

Finger dexterity will be measured by the Nine Hole Peg Test (9HPT) that is administered by asking the client to take pegs from a container, one by one, and to place them into holes on a board as quickly as possible; scores are based on the time taken to complete the test activity, recorded in seconds. Alternative scoring—the number of pegs placed within 50 or 100 s can be recorded. In this case, the results will be expressed as the number of pegs placed per second; a stopwatch should be started from the moment the participant touches the first peg until the moment the last peg hits the container.

Upper extremity proximal motor control and dexterity during ADL will be measured by the Frenchay Arm Test (FAT). The FAT is an upper-extremity-specific measure of activity limitation (activity domain of the ICF). Each item is scored as either pass (=1) or fail (=0). The total scores range from 0 to 5.

Performance in ADL will be measured by the modified Barthel Index (mBI), which is an ordinal scale used to measure ten variables describing ADL and mobility, with a higher number being a reflection of greater ability to function independently following hospital discharge. The 10 items of mobility and self-care ADL are feeding, personal hygiene, bathing, dressing, chair–bed transfer, toileting, bladder continence, bowel continence, ambulation or wheelchair use, and stair climbing. Scoring: the item scores are summed across them in order to compute the total score; a score of 0 indicates total assistance, while a total score of 100 indicates total independence.

The degree of disability or dependence during daily activities will be measured by the modified Rankin Scale (mRS), which is a commonly used scale for measuring disability or dependence in people who have suffered a stroke or other causes of neurological disability. It has become the most widely used clinical outcome measure for stroke clinical trials. The scale runs from 0 to 6, ranging from perfect health without symptoms to death: 0, no symptoms; 1, no significant disability—able to carry out all usual activities, despite some symptoms; 2, slight disability—able to look after own affairs without assistance, but unable to carry out all previous activities; 3, moderate disability—requires some help, but able to walk unassisted; 4, moderately severe disability—unable to attend to own bodily needs without assistance, and unable to walk unassisted; 5, severe disability—requires constant nursing care and attention, and is bedridden and incontinent; and 6—death.

Moreover, an instrumental robotic assessment through the exoskeleton will be carried out only on subjects in the EG:A_FORCE: measures of the force exerted by the patient for each movement;A_MOVE: measures the patient’s 3D work area (paint the walls of the room);A-GOAL: movement functionality.

### 2.6. Neurophysiological Signals

In a subgroup of subjects in the EG, changes in muscle activity will be assessed by surface ElectroMyoGraphy (sEMG) at T0 and T1 during the execution of a standardized motor task (reaching). Specifically, the muscle activity of the following muscles of both the healthy and paretic upper extremity will be acquired: pectoralis major, ascending trapezius, transverse trapezius, medial deltoid, biceps brachii, triceps brachii, flexor carpi ulnaris, and extensor carpi ulnaris. The electrodes should be placed halfway between the origin and insertion, parallel to the muscle fibers, according to SENIAM recommendations [18]. To analyze the amplitude of the surface EMG signal, we will calculate the root mean square, i.e., the square root of the average power of a signal over a given period of time. For this purpose, the raw EMG data will be full-wave rectified and processed using an algorithm with a sliding window of 20 ms. The EMG with the largest rectified and smoothed amplitude will be quantified for a 2 s period during each test. Data from this period will be used to analyze each muscle test performed for normalization.

Moreover, the electroencephalography (EEG) will be assessed as follows: brain electrical activity will be recorded (0.3–100 Hz bandpass, sampling frequency: 512 Hz) from 120 electrodes placed according to the International 10–20 System during resting conditions with the patients’ eyes open and closed (5 min each). A horizontal and vertical electroculogram (0.3–70 Hz bandpass) will be recorded to monitor eye movements. The EEG evaluation will be performed before and after rehabilitation. To eliminate interference from eye, muscle, and cardiac artifacts and other types of noise, the EEG will be fragmented into 2 s epochs, and two procedures will be used: data will be reviewed to manually eliminate epochs with aberrant waves; artifact detection will be completed using an independent component analysis (ICA) algorithm developed in EEGLAB. Private epochs of artifacts will be considered for subsequent analyzes. The frequency bands of interest are delta (2–4 Hz), theta (4–8 Hz), alpha1 (8–10.5 Hz), alpha2 (10.5–13 Hz), beta1 (13–20 Hz), beta2 (20–30 Hz), and gamma (30–40 Hz). For the EEG source analysis, the EEG data will be normalized, and the activation current density of the cortical sources on 6239 voxels will be calculated using standardized low-resolution electromagnetic tomography (sLORETA), available as a free software package. This method is based on the resolution of the inverse problem by reconstructing the cortical distribution of the sources of neuronal electrical activity in three dimensions, starting from the EEG data. Given the low spatial resolution of the method, 12 brain regions of interest (central, frontal, occipital, temporal, and limbic in the right and left hemispheres) will be reconstructed based on the Talairach atlas. An EEG spectral coherence analysis that evaluates functional coupling between the brain areas studied will be implemented using software developed in our laboratory (Matlab, Mathworks Inc., Natick, MA, USA). Brain connectivity will be calculated using eLORETA software for 84 regions of interest defined according to the 42 Brodmann areas for the right and left hemispheres. Using the 84 eLORETA regions of interest, delayed linear coherence will be calculated using the “all voxels closest” method among all possible pairs of regions of interest. The connectivity values will be calculated for each frequency band and for each subject and will be used as the weight of the graph calculated with graph theory. To study connectivity and to track its modulation (graph theory) after rehabilitation treatment, innovative brain network analysis measures will be used. The following parameters will be calculated for the sources of cerebral activation, for each participant, and in each frequency band: The characteristic path length (L) represents a measure of brain integration and is given by the average of the shortest path between each pair of connected nodes. The clustering coefficient (C) represents a measure of brain segregation, quantified as the tendency of the network to form clusters. Small-worldness (S) is the ratio between the normalized C and L.

## 3. Procedure

### 3.1. Control Group

The control group (CG), in addition to multidisciplinary rehabilitation, will follow 40 min of conventional upper limb rehabilitation. Each participant will perform a total of 25 ± 3 conventional upper limb treatment sessions with a frequency of 5 times a week for 5 weeks. Each session will consist of passive, active-assisted, and active exercises tailored for shoulder, arm, and hand motor rehabilitation. For a detailed description of the exercises scheduled, please see Table 1, which shows the Tidier Template for Intervention Description and Replication [19].

### 3.2. Experimental Group

The experimental group (EG), in addition to multidisciplinary rehabilitation, will perform one session per day of exoskeleton-assisted UL-RAT through the Armeo^®^Power (Hocoma AG, CH) robotic system. Each participant will perform a total of 25 ± 3 treatment sessions with a frequency of 5 times a week for 5 consecutive weeks.

During the first session, the device will be adjusted to the patient’s arm size and the angle of suspension. The working space and the exercises will be selected once the upper limb has been fitted with the system. The selection of personalized exercises will be based on the motor skills of each patient, and the difficulty can be gradually increased during training. In particular, a course of exercises has been defined, in which the difficulty varies (the suspension rate; the level of assistance; and the complexity of movement (1D, 2D, and 3D)). The physiotherapist will choose the modality based on the patient’s motor skills (standardized and personalized training); see Table 1.

UL–RAT will be set for 25% of the total weekly regular rehabilitation, in addition to the remaining 75% of tailored multidisciplinary rehabilitation programs, including occupational therapy, speech therapy, and cognitive-neuropsychological therapy. As compared with EG, CG will receive the same amount of neuro-motor rehabilitation each day following traditional rehabilitation approaches.

### 3.3. Safety and Adverse Event Reporting

All safety issues concerning the use of robotics will be applied for this trial. Moreover, the main investigators will report and evaluate suspected adverse events during the training and at follow-up visit.

### 3.4. Data Management

A long-term data sharing and preservation plan will be used to store and make the data publicly accessible beyond the life of the project. The data will be deposited into an online data repository. Before any testing under this protocol, the subjects will also provide all authorizations required by local law (D.Lgs. 196/2003). Each subject will be identified by a code in an unequivocal manner, which will be the identifier of the subject for the duration of the study.

### 3.5. Data Analysis

The sample size was determined based on the data published by Calabrò et al. in 2017 [20]: using a statistical power of 80% and an alpha of 5% with a β error of 20%, we need 70 total subjects (35 EG and 35 CG). However, considering the potential drop-out rate, 20% more, i.e., a total of 84 patients (42 EG and 42 CG), should be enrolled. In the absence of standard values of reference in the literature for the evaluation of the neurophysiological effects of robotic treatment for an upper limb compared with conventional treatment, an interim analysis will be performed when we enroll at least 10 patients of the expected sample. Following the results of the analysis of the acquired neurophysiological signals, we will evaluate whether to continue the enrolment or re-evaluate the number of subjects to be recruited.

## 4. Discussion

This multicenter, single-blinded RCT represents the first attempt to prove the superiority of a 3D task-specific UL-RAT delivered by a powered exoskeleton over time-matched, conventional arm rehabilitation in a large cohort of subacute stroke patients. The first weeks after a stroke are a critical time window for motor recovery [21], and the role of rehabilitation in this phase is to promote neural repair and the optimization of functioning. It has been shown that early, intensive, repetitive, and task-oriented training may lead to better functional recovery, and such training can be better provided with a robot than with conventional physiotherapy. In fact, robotic rehabilitation can be applied in the very early phases of the disease, and the robot can help perform more intensive, repetitive, and reproducible training. This is why patients who may benefit more from robotics are those more severely affected and within the first months after the event [2].

The expected results of the present multicenter RCT will be (a) to achieve better motor recovery in terms of motor functioning following UL-RAT through the exoskeleton system, thus obtaining greater autonomy in the performance of the ADL and better participation; (b) to identify a customizable rehabilitation protocol for upper limb motor recovery based on the clinical characteristics of the patient (in order to move toward personalized medicine); (c) to classify the clinical and neurophysiological characteristics for the prediction of rehabilitation outcomes in terms of motor recovery; and (d) to investigate if the acquired rehabilitative effects might also be maintained at follow-up.

Concerning the second point, it is conceivable that the better recovery expected in patients receiving the UL-RAT might be related to both the use of the UL exoskeleton and the 2D non-immersive virtual reality during the training [22].

The study design represents a strength of the research. In fact, a stratification of the sample in blocks by time since the event and by severity ensures that the groups are not biased by such determinant variables. In addition, the method used allows us to understand the influence of stroke time and severity on the efficacy of robot-assisted therapy. Future research should focus, as also underlined by the Cochrane revision [2], on the effect of dose and frequency of robotic therapy on the primary outcome, designing pragmatic trials specifically for these objectives (and not verifying them as secondary objectives).

Indeed, it has been shown that visual feedback leads to greater involvement, motivation, and participation of the patients with better functional outcomes. Non-immersive virtual reality has a positive impact not only on motor recovery but also on cognitive function and, therefore, on mood and behavior. Neural plasticity is boosted by virtual reality thanks to the task-oriented exercises, as well as the visual and acoustic feedback that provides the patients with knowledge about both their performance and their results, further favoring reinforcement learning [23,24,25].

In this protocol, the neuroplasticity changes underpinning motor recovery will be assessed in terms of the use of EEG. In fact, as demonstrated for the lower limb, the tool is capable of detecting specific markers (e.g., premotor-parietooccipital desynchronization of γ-oscillations) of the activation of sensorimotor and visuo-spatial associative areas concerning motor planning and selective muscle activation [26]. Moreover, a specific EEG analysis may be used as a predictor of UL post-stroke functional recovery [15,26].

This protocol has some limitations. Firstly, spontaneous recovery will be the main determinant for the trajectories of the obtained recovery. Therefore, it will be difficult to separate the observed time-dependent changes over time, discriminating how much they are due to biological processes or the rehabilitation interventions and environments. To overcome this specific issue, a stratification for time since stroke has been planned. Secondly, the neurophysiological and kinematic assessment will be conducted in a subgroup of patients, preventing us from drawing definite conclusions on the differences among treatments on the brain dynamics and quality of arm movements.

## 5. Conclusions

This RCT might shed some light onto the real efficacy of UL-RAT in patients with stroke, as compared with conventional treatments. Since it will investigate the neurophysiological basis underpinning functional recovery using a combined sEMG-EEG approach, the results could be useful in justifying a wider use of exoskeletons in clinical practice.

## Figures and Tables

**Figure 1 brainsci-13-00700-f001:**
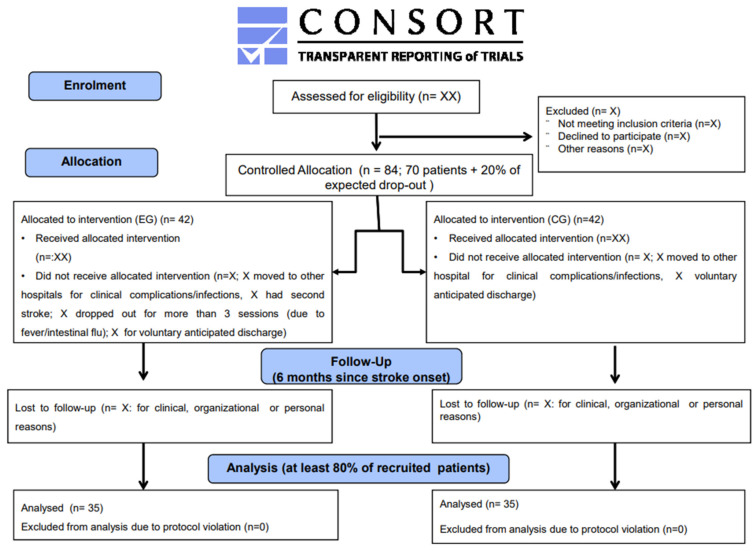
Consort flow diagram.

**Table 1 brainsci-13-00700-t001:** Overview of an intervention session in the experimental and control groups according to the Template for Intervention Description and Replication (TIDieR) checklist and guide. Both groups will receive 25 sessions of rehabilitation treatment for three weeks (5 sessions/week).

TIDieR Item	Experimental Group	Control Group	Experimental and Control Groups			
1. Name	Upper limb rehabilitation through Armeo Power exoskeleton	Upper limb rehabilitation through conventional therapy	Passive, active-assisted, and active exercises addressed for shoulder, arm, and hand motor rehabilitation	Exercises for muscle strength	Stretching exercises	Functional activity training
2. Why	To improve upper limb motor functioning	To improve upper limb motor functioning	To improve mobility and muscle activation, to preserve the passive range-of-motion (ROM), and to prevent contractures and spasticity	To improve muscle strength	To prevent muscle retraction/contractures and spasticity	To improve autonomy in activities of daily living
3. What (materials)	Exoskeleton with auditory and visual bio-feedback	Proprioceptive and therapy objects	With or without continuous shoulder and/or elbow passive motion	Soft weights, elastic resistance bands, and therapy objects assisted by physiotherapist	Manually assisted by physiotherapist	Objects typically present in the home for activities of daily living
4.What (procedures)	The exoskeleton will be donned and customized on the patient and they will be asked to interact with it in an assist-as-needed modality, receiving a real-time visual biofeedback and interacting with serious video-games.During the first session, the device will be adjusted to the patient’s arm size and the angle of suspension. The working space and the exercises will be selected once the upper limb has been fitted with the system. The selection of personalized exercises is based on the motor skills of each patient and the difficulty can be gradually increased during training.	The exercises will consist of passive, active-assisted, and active exercises for the shoulder, elbow, and wrist in a seated position	Gradual increasing the degree of flexion and extension or a physical therapist will passively mobilize the joints into flexion and extension	The exercises will consist of isometric, isotonic contraction of the upper limb muscles	The exercises will consist of stretching the upper limb muscles	The patients will follow training on activities of daily living. The patient will perform training with assistance, with aids, or autonomously.
5. Who provided	Senior physical therapists who are experts in neurorehabilitation					Occupational Therapists
6. How	An individual face-to-face treatment session					
7. Where	Rehabilitation gym in an intensive subacute rehabilitation hospital					
8. When and how much	45 + 60 (occupational therapy; functional therapy (trunk control, standing, and walking training), speech therapy, and/or neuro-cognitive therapy)					
9. Tailoring			The exercise will be tailored to participants’ goals, current abilities, and preferences, also considering the patient’s point of view			
10. Modifications	The exercises difficulty will be increased gradually session by session, depending on current progress, level of pain, and ability.					
11. How well (planned)	Organizing periodical updating meetings with the team of physical therapists and other clinicians involved in the study to conduct the treatments in the most homogeneous way possible and to address any problems and/or critical points that may be encountered during the trial execution.					
12. How well (actual)	Physical therapist delivering the treatments will register how many sessions a patient attended and completed.					

Abbreviations: TIDieR: Template for Intervention Description and Replication.

## Data Availability

Not applicable.

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
