# Peer review of "Neurophysiological and Clinical Effects of Upper Limb Robot-Assisted Rehabilitation on Motor Recovery in Patients with Subacute Stroke: A Multicenter Randomized Controlled Trial Study Protocol"

_brainsci, 2023, doi:10.3390/brainsci13040700_

Round 1
Reviewer 1 Report
The manuscript introduces the reader to a protocol for a planned study on the impact of robotic upper limb rehabilitation on motor recovery in patients with subacute stroke. As indicators of recovery, it is planned to use a number of used methods of clinical evaluation of functions, as well as neurophysiological criteria. Since we are not going to discuss the original results, we note some features that should be taken into account.
1. The criterion for verifying the final diagnosis of patients selected for the study is not clear. Traditionally, the final diagnosis is based on MRI. To some extent, this criterion should be taken into account in the classification of selected patients. This will help to solve the problems that the authors form.
2. The authors plan to use traditional neurophysiological methods - EMG and EEG. But it is necessary to choose processing methods already at the stage of the protocol, especially for EEG. A lot of EEG processing methods exist, even in eLORETA. What data do the authors want to receive? This should be justified with the choice of EEG processing. If it is possible to select somatosensory evoked potentials at the protocol stage, the study will be re-scored. But it doesn't have to be here.
3. It is not easy to draw up a conclusion without results. The authors make an attempt to substantiate the advantages of a combined study. It is advisable to evaluate the methods used step by step and take a step into future research.
4. Add to the list of references works on neurophysiological research after stroke in neurology and neurosurgery, and add relevant information to the introduction and discussion.
Author Response
The manuscript introduces the reader to a protocol for a planned study on the impact of robotic upper limb rehabilitation on motor recovery in patients with subacute stroke. As indicators of recovery, it is planned to use a number of used methods of clinical evaluation of functions, as well as neurophysiological criteria. Since we are not going to discuss the original results, we note some features that should be taken into account.
Thank you for the revision. We changed the manuscript in accordance with your suggestions.
- The criterion for verifying the final diagnosis of patients selected for the study is not clear. Traditionally, the final diagnosis is based on MRI. To some extent, this criterion should be taken into account in the classification of selected patients. This will help to solve the problems that the authors form.
Thanks for the opportunity to clarify this point. We added “verified by brain imaging (CT or MRI)” to the method section.
- The authors plan to use traditional neurophysiological methods - EMG and EEG. But it is necessary to choose processing methods already at the stage of the protocol, especially for EEG. A lot of EEG processing methods exist, even in eLORETA. What data do the authors want to receive? This should be justified with the choice of EEG processing. If it is possible to select somatosensory evoked potentials at the protocol stage, the study will be re-scored. But it doesn't have to be here.
Thank you for the instructive suggestion. We have added further details on the processing of neurophysiological data in this protocol.
- It is not easy to draw up a conclusion without results. The authors make an attempt to substantiate the advantages of a combined study. It is advisable to evaluate the methods used step by step and take a step into future research.
We would thank the reviewer for the suggestion: we have added a new paragraph on this aspect in the discussion section.
- Add to the list of references works on neurophysiological research after stroke in neurology and neurosurgery, and add relevant information to the introduction and discussion.
This data on the role of EEG as predictor of outcomes in neurology and neurosurgery have been added.
Reviewer 2 Report
I would like to congratulate the authors of this project. It is a well-designed protocol and one that I hope can be properly implemented in practice. After reviewing the manuscript, I would like to point out some modifications that I believe should be indicated:
1. In the description of the inclusion criteria, they state the following: “sufficient cognitive and linguistic level to understand the instructions and provide informed consent to participate in the study”. I think it would be appropriate to use a cognitive screening tool to ensure that this condition is met, due to the cognitive and executive function alterations observed in stroke patients. Have you considered using a test such as the MoCA?
2. In the description of the exclusion criteria they indicate the following: “interruption of treatment for 1 week or 5 consecutive sessions”.I think you should specify that you refer to the treatment raised in this protocol and not to an conventional therapy intervention. In case they were patients attending physiotherapy or occupational therapy rehabilitation, has this been considered an exclusion criterion?
3. Regarding the recruitment of the sample, it is not specified from where it will be obtained. I think it would be appropriate to specify this a little more in this section.
4. Please review the figure as there are several words that are underlined in blue and red: "n=XX; drop-out; analysed".
5. In the case of the description of the control group intervention, I understand from what you have detailed, that this group does not receive occupational therapy, speech therapy or neuropsychology intervention. Don't you think that this could be a bias when comparing the results with the experimental group, which does receive this intervention? It is true that neuropsychology or speech therapy does not treat the upper limb, but occupational therapists do. How did you plan to analyze this?
6. The sentence on lines 268-269 is not well written, please revise it: "[...]the analysis will be performed when 10 patients of the expected sample have arrived".
7. The sentence on line 264 needs to be revised. There is a grammatical error.
8. One of the objectives stated in the discussion section is the following: to achieve better motor recovery both in terms of the range of motion and muscle recruitment associated with cognitive stimulation following the UL-RAT through the exoskeleton system. In the evaluation tests section they have not included an objective measurement of joint range. The Modified Ashworth Scale scores refer to improvements in relation to muscle activation and tone, however, they do not objectively measure joint range. Have you considered including a goniometric measurement prior to the start of the intervention?
Author Response
I would like to congratulate the authors of this project. It is a well-designed protocol and one that I hope can be properly implemented in practice. After reviewing the manuscript, I would like to point out some modifications that I believe should be indicated:
Thank you for the revision. We changed the manuscript in accordance with your suggestions.
- In the description of the inclusion criteria, they state the following: “sufficient cognitive and linguistic level to understand the instructions and provide informed consent to participate in the study”. I think it would be appropriate to use a cognitive screening tool to ensure that this condition is met, due to the cognitive and executive function alterations observed in stroke patients. Have you considered using a test such as the MoCA?
Thanks to the reviewer for this comment. We decided to not include general cognitive tests that can be very difficult to perform during the screening session. However, the presence of an attention or linguistic deficits that would not interfere with the ability to conduct the study will be tracked for post-hoc analysis.
- In the description of the exclusion criteria they indicate the following: “interruption of treatment for 1 week or 5 consecutive sessions”.I think you should specify that you refer to the treatment raised in this protocol and not to an conventional therapy intervention. In case they were patients attending physiotherapy or occupational therapy rehabilitation, has this been considered an exclusion criterion?
We amended. We have specified that an interruption of both treatments will be considered as an exclusion criteria.
- Regarding the recruitment of the sample, it is not specified from where it will be obtained. I think it would be appropriate to specify this a little more in this section.
Thanks for this comment. We included a sentence related to the center that will be involved in the recruitment.
4.Please review the figure as there are several words that are underlined in blue and red: "n=XX; drop-out; analysed".
Thank you, corrected.
5.In the case of the description of the control group intervention, I understand from what you have detailed, that this group does not receive occupational therapy, speech therapy or neuropsychology intervention. Don't you think that this could be a bias when comparing the results with the experimental group, which does receive this intervention? It is true that neuropsychology or speech therapy does not treat the upper limb, but occupational therapists do. How did you plan to analyze this?
Thanks for this comment. Patients will receive in both groups a multidisciplinary rehabilitation according to their rehabilitation needs. The UL rehabilitation will be delivered as conventional therapy (CG) and as conventional therapy + UL-RAT (EG). We can assume that all the other therapies that might interfere with arm recovery (except to conventional therapy or RAT) will be balanced among groups with the randomization.
- The sentence on lines 268-269 is not well written, please revise it: "[...]the analysis will be performed when 10 patients of the expected sample have arrived".
This sentence has been clarified.
7.The sentence on line 264 needs to be revised. There is a grammatical error.
Corrected.
8.One of the objectives stated in the discussion section is the following: to achieve better motor recovery both in terms of the range of motion and muscle recruitment associated with cognitive stimulation following the UL-RAT through the exoskeleton system. In the evaluation tests section they have not included an objective measurement of joint range. The Modified Ashworth Scale scores refer to improvements in relation to muscle activation and tone, however, they do not objectively measure joint range. Have you considered including a goniometric measurement prior to the start of the intervention?
Thank you for the comment. It was our mistake. The main objective of the study is “to achieve better motor recovery in terms of motor functioning following the UL-RAT through the exoskeleton system” since the primary outcome of the study is the Fugl-Meyer Assessment of the Upper limb.